

# Frequency and quality of first aid offered by older adolescents: a cluster randomised crossover trial of school-based first aid courses

Alyssia Rossetto[1,2], Amy J. Morgan[1], Laura M. Hart[1,3], Claire M. Kelly[2] and Anthony F. Jorm[1,2]

[1] Melbourne School of Population and Global Health, The University of Melbourne, Melbourne, VIC, Australia
[2] Mental Health First Aid Australia, Melbourne, VIC, Australia
[3] School of Psychology and Public Health, La Trobe University, Melbourne, VIC, Australia

Corresponding author
Alyssia Rossetto,
alyssia.rossetto@unimelb.edu.au

## ABSTRACT

**Background:** Research indicates that school-based first aid programmes appear to improve students' knowledge and skills. However, evidence for their effectiveness is limited by a lack of rigorously designed studies. This research used a cluster randomised crossover trial to assess the effects of two different types of first aid training on the frequency and appropriateness of older adolescents' first aid behaviours towards their peers 12 months after training.
**Methods:** Schools eligible to participate were government funded and able to accommodate first aid training and survey time for two consecutive Year 10 student cohorts. Four Australian public schools were matched in two pairs and randomly assigned to receive either physical first aid (PFA) or teen mental health first aid (tMHFA) training for their Year 10 student cohort (mean age 16 years). In the second year, the new Year 10 cohort received the other intervention. Four cohorts were randomised to receive PFA and four were randomised to receive tMHFA. Online surveys were administered at baseline and 12 months after training, measuring whether students had encountered a peer needing PFA, whether they had provided PFA, what actions they performed and, if applicable, why they had been unable to help the person. Only research staff analysing the data could be blinded to measurement occasion, school identity and condition.
**Results:** Four cohorts received PFA and four received tMHFA. The results indicated that there were no differences between groups regarding the frequency of appropriate first aid actions performed 12 months after training. The most common types of PFA provided to a peer were sending for help and wound care. Students most commonly mentioned someone else attending to their peer or lacking skills or experience as reasons for not performing PFA actions.
**Conclusions:** More research that examines first aid behaviours using rigorous, longitudinal study designs is needed to establish the effectiveness of school-based first aid training for older adolescents.

# INTRODUCTION

The ability to provide first aid is an important set of skills for anyone of any age to have. The provision of appropriate first aid to an ill or injured person can save a life, mitigate the severe consequences of sudden illness or injury, address minor injuries in a timely and effective way, and create a caring and reassuring environment until emergency services arrive.

First aid skills may be particularly important for groups who are often exposed to illness and injury, such as children and adolescents (*Wilks & Pendergast, 2017*). Rates of hospital admissions for conditions such as anaphylaxis are rising, particularly in the 5–14 year age group (*Mullins, Dear & Tang, 2015*). Accidents and injuries are a leading cause of hospitalisation, morbidity and mortality among children and adolescents in Australia and worldwide (*Australian Institute of Health & Welfare, 2018*; *Global Burden of Disease Pediatrics Collaboration, 2016*; *Pointer, 2014*). In Australia, the most common mechanisms of injury for young people aged between 11 and 17 are falls, road transport and intentional self-harm (*Mitchell, Curtis & Foster, 2018*; *Pointer, 2014*). Transport-related injuries, collisions and falls are also the most common causes of major paediatric trauma in 11–15 year olds in the state of Victoria, and these injuries primarily occur in settings outside the home and school (e.g. on roads, streets and highways, in recreation areas, and in athletics and sports areas; *Beck et al., 2019*). This suggests that illnesses and injuries in young people are common and take place in areas where bystanders, such as peers, may be present and able to offer first aid.

One way to increase the proportion of young people equipped with first aid skills is to offer first aid training to students in schools. Schools have an influential role in promoting adolescent health and wellbeing, and provide access to the majority of young people from diverse socio-economic backgrounds. Although first aid training for students is not currently mandatory in Australia, schools appear to be supportive of first aid courses for students, and first aid principles align well with Australian school curriculum goals of developing students' knowledge, skills, self-efficacy and resilience across various life domains (*Wilks & Pendergast, 2017*).

Many studies have assessed the effectiveness of school-based first aid programmes. Systematic reviews indicate that first aid training appears to improve students' knowledge and skills in areas such as injury prevention, cardiopulmonary resuscitation (CPR) and use of automatic external defibrillators (*Plant & Taylor, 2013*; *Reveruzzi, Buckley & Sheehan, 2016*). However, existing evidence for the effectiveness of school-based first aid programmes is limited by a lack of well-designed studies. Common methodological issues include a lack of random allocation, absence of comparison or control groups, small or biased samples, failure to measure students' first aid knowledge or experiences prior to completing school-based first aid training, and a lack of justification for the topics addressed in the training (*Reveruzzi, Buckley & Sheehan, 2016*). These issues indicate that
more rigorous study designs are needed to examine the acquisition and retention of first aid skills in adolescent students undertaking school-based first aid training. Studies that assess whether first aid skills are used, and used appropriately, outside of learning and skills testing settings are also needed to understand whether and how adolescents apply first aid when faced with a real-life emergency.

The current article reports on a subset of results from a randomised controlled trial which compared a physical first aid (PFA) course with an equivalent length mental health first aid course (teen Mental Health First Aid; tMHFA) to assess the effects of each type of training on older adolescents' PFA behaviours 12 months after the training. This study aimed to establish whether the training programmes produced differences in the frequency with which students offered and provided PFA to peers experiencing an illness or injury and the amount of first aid provided that was in line with current PFA guidelines. It was hypothesised that students receiving the PFA course would offer PFA more often, provide PFA more often, and provide more appropriate PFA than students receiving the tMHFA course. The study also aimed to ascertain which PFA actions were most commonly performed amongst this age group and why first aid might not have been provided in situations where a peer required PFA. Lastly, the study aimed to assess predictors of appropriate PFA.

## MATERIALS AND METHODS

The description below outlines the sections of the trial design and study methodology relevant to the aims of this research. It conforms to the CONSORT 2010 statement and extension for cluster randomised trials (*Campbell et al., 2012*). Full details of the trial and methodology can be found in *Hart et al. (2018)*.

### Study design

A cluster randomised crossover trial with four schools was conducted between 2014 and 2017. Each school comprised one cluster and provided two cohorts of Year 10 students during the study. Each school received both interventions, with one student cohort completing one intervention in the first year, and a second student cohort completing the other intervention in the second year. Schools enrolled in the trial were pair matched based on their Index of Community Socio-Educational Advantage (ICSEA) and the size of their Year 10 cohort. Pairs were then randomly assigned to receive either tMHFA or PFA for their Year 10 students in the first year of the trial. In the second year, the new Year 10 students received the other intervention. This trial design reduced the number of clusters required to achieve adequate participant numbers, enabled counterbalancing across schools, and allowed analyses to control for within-school variance.

### Participants and procedure

Schools in the Melbourne region which had expressed an interest in receiving mental health first aid training for staff and students were contacted to discuss the research. Schools eligible to participate were government funded and able to accommodate three 30-min survey sessions and three 75-min training sessions during regular class time for

two consecutive Year 10 student cohorts. Upon completion of a research agreement signed by the school principal, the research team enrolled and matched schools in the trial. The intervention sequence for each pair of schools (i.e. whether schools received tMHFA or PFA first) was determined by the trial manager using a random sequence generator. The trial's research assistant allocated the first sequence to the first school enrolled in the trial (and its pair), and allocated the second sequence to the second school (and its pair). Researchers, instructors, schools and students could not be blinded to intervention type. Survey and training sessions were then scheduled, using information provided by schools regarding the number of Year 10 students and classes in their cohorts.

All students in each cohort were eligible to complete evaluation surveys if they had parental consent and provided assent by selecting a checkbox that confirmed they wished to proceed at the beginning of each survey. Students, parents and teachers were informed about the research through information sessions, electronic and hard copy information forms, and school-based communications (e.g. newsletters). Opt-out (passive) parental consent was used; parents could opt their child out of the training and/or surveys at any time by returning a signed form to the school.

All surveys were hosted on the online platform Survey Monkey. Schools were emailed a link to each survey, which was distributed to students through an intranet or student email address. Students entered a unique participant identification number at the beginning of each survey, which enabled their responses to be matched over time but did not permit them to be identified by the researchers. Students completed a baseline survey 1 week prior to the first training session, a post-course survey up to 1 week after the final training session, and a follow-up survey 12 months after the final training session.

All students were eligible to take part in the training allocated to their cohort if they had parental consent. The PFA and tMHFA training programmes each consisted of three 75-min classroom-based sessions, with sessions typically beginning 1 week after completion of the baseline survey and delivered 1 week apart. Each programme was presented by trained external instructors. Table 1 summarises the content of each programme by session. The content covered appropriate responses to health problems that adolescents were likely to encounter in their peers. PFA content was delivered by accredited St. John Ambulance and Red Cross instructors through didactic instruction, group discussion, demonstrations and practical activities with mannequins, bandages and splints. The course was developed for this research and was shorter in duration and more limited in content than accredited first aid courses offered in Australia. tMHFA is a skills-based programme that teaches adolescents how to recognise and respond to a peer experiencing mental ill health. Courses were delivered by accredited tMHFA instructors through a Powerpoint presentation, films, role-play activities, group discussion, small group activities, a manual and workbook activities (Hart et al., 2016). Students in both interventions received a certificate of completion at the end of their training.

## Measures

The questions described below were administered during the baseline and 12-month follow-up surveys. Post-course surveys did not include these questions as it was unlikely

**Table 1 Structure and content of first aid interventions.**

| Session number and duration | PFA | tMHFA |
|---|---|---|
| Session 1 (75 min) | First aid and the DRSABCD action plan<br>– Where to start<br>– When and how to call emergency services<br>– How to manage a conscious person<br>– How to manage an unconscious person<br>– When and how to perform cardiopulmonary resuscitation<br>– When and how to use an automated external defibrillator | Mental health and mental health problems<br>– What is mental health?<br>– What are mental health problems?<br>– Types of mental health problems<br>– Impact on young people<br>– What is stigma?<br>– Types of appropriate help |
| Session 2 (75 min) | Basic first aid for:<br>– Sprains<br>– Strains<br>– Wound care<br>– Fractures and dislocations<br>– Concussion<br>– Asthma | Helping a friend in a mental health crisis<br>– What is mental health first aid?<br>– What is a mental health crisis?<br>– Using the tMHFA action plan to help a friend in crisis<br>– Recovery position |
| Session 3 (75 min) | Basic first aid for:<br>– Anaphylaxis<br>– Poisons<br>– Exposure to heat<br>– Exposure to cold<br>– Diabetes<br>– Seizure | Helping a friend who is developing a mental health problem<br>– The importance of acting early<br>– Using the tMHFA action plan to help a friend in crisis<br>– Helpful links and resources |
| Action plan | D: Danger<br>R: Response<br>S: Send for help<br>A: Open airway<br>B: Check for breathing<br>C: Start CPR<br>D: Attach defibrillator | Look: Look for warning signs<br>Ask: Ask how they are<br>Listen: Listen up<br>Help: Help them connect with an adult<br>Your Friend: Your friendship is important |

**Note:**
PFA, Physical First Aid; tMHFA, teen Mental Health First Aid.

that students would have had an opportunity to administer first aid in the time between completing the training and beginning the post-course survey. As there are no standardised scales in this area of research, questions from the First Aid Experiences Questionnaire (*Hart et al., 2019*, *2016*; *Jorm, Kitchener & Mugford, 2005*) were adapted to ask respondents about PFA situations they had encountered.

Students firstly completed two sections based on two vignettes of adolescents experiencing mental health problems (data reported elsewhere; see *Hart et al. (2016)* and *Hart et al. (2018)*). The next section asked 'In the last 12 months (baseline)/Since completing the first aid course 12 months ago (follow-up), have you come across someone about your age (i.e. between 13 and 18 years old) who has required first aid because of an

emergency or injury?' Students who responded 'No' or 'I don't want to answer this question' moved on to the next section of the survey, while students who answered 'Yes' or 'Not sure' proceeded to answer additional questions about their experiences.

Students were asked to type in the number of people about their age they had encountered who required assistance for an emergency or injury, and then asked whether they had offered any help to the person they knew best. Students who responded 'No' were asked why they had been unable to help the person (free text response), and students who answered 'Yes' or 'Not sure' were asked to describe what they did to help the person (free text response).

Students provided demographic information (age, gender, English as first language) at baseline, and indicated how many training sessions they attended (0, 1, 2 or 3) at 12-month follow-up.

## Sample size calculation

Sample size calculations conservatively estimated 100 Year 10 students per school, with 50% completing pre-intervention and 12-month follow-up surveys. This would result in 200 students per intervention ($n = 800$ students in total). The estimated intraclass correlation for students at the school cluster level was 0.003, based on findings from previous research (*Hart et al., 2016*). The intraclass correlation was not included in the power calculations because participating schools were matched and its effect on the study design was therefore likely to be limited. The study assumed a 0.70 correlation between pre- and post-intervention measurements, resulting in 0.80 power to detect small ($d = 0.17$) group-by-measurement occasion differences at $\alpha = 0.05$.

## Statistical analysis

Primary research outcomes were analysed at the individual level. Although this trial used cluster randomisation, this research aimed to assess the effect of each type of first aid training on students' subsequent behaviour towards their peers, rather than the effects of the training on the whole school community. Research staff analysing the data were blinded to measurement occasion, school identity and condition.

Free text responses describing first aid provided to a peer were analysed using a coding framework developed for this study, which was based on current PFA guidelines (https://resus.org.au/guidelines/) and content analysis approaches (*Crowe, Inder & Porter, 2015*). Each code corresponded to a topic discussed in the PFA sessions (e.g. check airway, asthma, wound care) and included instructions, examples and counter-examples to guide coder fidelity. Three members of the research team developed the initial coding frame, which was refined through discussion and double-coding of sample responses (selected to represent different codes). Coding discrepancies were discussed until consensus was reached. After finalising the coding framework, the remaining responses were coded. Responses were assigned a value of one if they corresponded to a first aid action covered in the training. Responses which were incorrect, inappropriate, or did not relate to any other codes in the coding frame were coded as zero. Responses could be coded into more than one category. Researchers coding and analysing data were blinded to

**Table 2 Participant numbers and analyses of first aid offered and provided by intervention.**

| Description | tMHFA | PFA | Total | Significance |
|---|---|---|---|---|
| Sample size at intervention allocation | 989 | 953 | 1,942 | |
| **Baseline data** | | | | |
| Number of surveys submitted | 821 | 803 | 1,624 | |
| Participation rate | 83% | 84% | 83.5% | |
| Number of students responding 'Yes' or 'Not sure' to the question 'In the last 12 months have you come across someone about your age who has required first aid because of an emergency or injury?' | 404 | 394 | 798 | $X^2(1) = 0.039$, $p = 0.844$, $V = 0.01$ |
| Number of students responding 'Yes' or 'Not sure' to the question 'Did you offer help to the person you knew best?' | 346 (85.6%) | 339 (86.0%) | 685 (85.8%) | $X^2(1) = 0.020$, $p = 0.887$, $V = 0.01$ |
| Number of students providing appropriate PFA to a peer | 38/260 responses (14.6%) | 80/272 responses (29.4%) | 118/532 responses (22.2%) | $X^2(1) = 16.861$, $p = 0.000$, $V = 0.18$ |
| **12-month follow-up data** | | | | |
| Number of surveys submitted | 465 | 429 | 894 | |
| Participation rate (as % of baseline surveys submitted) | 56.6% | 53.4% | 55.0% | |
| Number of students responding 'Yes' or 'Not sure' to the question 'In the last 12 months have you come across someone about your age who has required first aid because of an emergency or injury?' | 154 | 129 | 283 | $X^2(1) = 1.373$, $p = 0.241$, $V = 0.04$ |
| Number of students responding 'Yes' or 'Not sure' to the question 'Did you offer help to the person you knew best?' | 133 (86.4%) | 109 (84.5%) | 242 (85.5%) | $X^2(1) = 0.282$, $p = 0.595$, $V = 0.03$ |
| Number of students providing appropriate PFA to a peer | 16/96 responses (16.7%) | 27/85 responses (31.8%) | 43/181 responses (23.8%) | $X^2(1) = 5.674$, $p = 0.017$, $V = 0.18$ |

measurement occasion, school identity and intervention. Interrater reliability between coders was kappa = 0.88, $p < 0.001$. This coding frame is available as a Supplemental File.

Free text responses describing reasons for not providing first aid were analysed using a second coding framework, which was developed through content analysis of the open-ended responses to this question (*Crowe, Inder & Porter, 2015*). Two members of the research team developed the initial coding frame which was refined through discussion. One researcher then coded all responses. Responses which could not be interpreted, did not make sense or required the coder to make strong assumptions about the student's meaning were categorised as 'unable to code' and excluded from subsequent analyses. This coding frame is available as a Supplemental File.

Data were analysed using percent frequencies, chi-square analyses (using Cramer's V as a measure of effect size) and logistic regression. The intraclass correlation between appropriate first aid given and school cohort was 0.03, suggesting that school-level clustering had little influence on the data. Analyses were conducted using SPSS 25.

## RESULTS

Table 2 shows the number of students allocated to each intervention, participant numbers at each time point and data on the number of students who encountered a person needing

PFA, offered first aid, and provided appropriate first aid. The mean age of students completing the baseline survey was 15.87 years (SD = 0.52); 44.7% were female and 72.5% spoke English as a first language (see *Hart et al. (2018)* for full details). Of the students responding to the section on experiences of first aid, the mean age was 15.87 years (SD = 0.51), 46.4% were female and 76.6% spoke English as a first language. Chi-square analyses indicated that at both baseline and follow-up, there were no significant differences between interventions regarding the number of students who had encountered a person needing PFA in the last 12 months, or in the proportion of students who offered help to the person they knew best. However, at both time points, the group receiving PFA training provided significantly more appropriate first aid than the group receiving tMHFA. Logistic regressions that assessed intervention × time interactions for appropriate provision of PFA (controlling for age, gender, English as first language, school, intervention received, number of first aid sessions attended and whether appropriate PFA was provided at baseline) confirmed that appropriate PFA provided at baseline was the only significant predictor of appropriate help at each time point. Receipt of PFA training did not appear to improve the frequency or quality of students' first aid actions.

Of the total number of open-ended responses describing first aid provided to a peer, 60.3% were categorised as inadequate or ineffective. These responses included: talking to, calming or comforting the person with no mention of other actions taken; taking a peer to the school nurse or sick bay; offering assistance after the emergency had passed (e.g. 'Offered to carry person's books whilst their leg was in a moon boot'); and responses that did not describe a specific first aid action (e.g. 'Asked if there was anything I could do for them', 'Assist them in any way possible to make sure they were okay'). Table 3 summarises the most common types of appropriate PFA provided to a peer across both time points.

Table 4 summarises reasons given for not providing first aid at both time points. The most common reasons were because someone else was already providing first aid, because the student perceived their skill set or experience to be inadequate and because practical constraints precluded the provision of first aid.

## DISCUSSION

The results of this study indicate that brief PFA training does not appear to affect the frequency of appropriate first aid actions performed by older adolescents 12 months later. Despite significant differences between the PFA and tMHFA groups at follow-up, pre-existing differences at baseline suggested that the programme was not effective at changing behaviour. A lack of improvement in, or retention of, adolescents' first aid skills at 12 month follow-up is an atypical finding in the literature on school-based first aid programmes (*Moore, Plotnikoff & Preston, 1992*). One explanation for this finding may be that the bespoke, shortened PFA course used in this research may have reduced effectiveness when compared with lengthier, accredited PFA programmes, although at just under 4 hours duration it is somewhat longer than other first aid courses assessed in school settings (*Reveruzzi, Buckley & Sheehan, 2016*). This finding may also have resulted

**Table 3 Most common types of first aid provided to a peer with a physical injury or illness.**

| First aid response (n) | Example quote | Percentage of all responses (n = 916) | Percentage of appropriate responses (n = 172) | n (%) of appropriate responses from PFA group | n (%) of appropriate responses from tMHFA group |
|---|---|---|---|---|---|
| Send for help (52) | 'I asked if they required any medical attention and rang 000' (P1612, baseline) | 5.7 | 30.4 | 33 (63.5) | 19 (36.5) |
| Wound care (48) | 'Well it was my nephew and he had a deep cut on his head, I first got a towel and put water on it and told him to hold it against the wound, and as it was still bleeding and no one was home—I dialled 000 to ask for help and it went on from there.' (P74, follow-up) | 5.2 | 28.1 | 34 (70.8) | 14 (29.2) |
| Strains and sprains (26) | 'Performed RICER' (P992, follow-up) | 2.8 | 15.2 | 13 (50.0) | 13 (50.0) |
| Fractures and dislocations (13) | 'Made a sling out of my T-shirt and carried him to a safe place' (P976, baseline) | 1.4 | 7.6 | 11 (84.6) | 2 (15.4) |

Note:
A total of 676 responses were recorded at baseline and 240 at follow-up. Only categories with responses provided by more than five participants are included in this table. Other responses, reported by between one and five participants, related to: checking for danger, DRSABCD, checking airways, performing CPR, asthma management, seizure management, allergy management, fainting management, spinal injury management, checking for a response, concussion management and anaphylaxis management. Logistic regressions that assessed whether type of PFA provided at baseline predicted the same type of PFA at follow-up (controlling for age, gender, English as first language, school, intervention received and number of first aid sessions attended) were non-significant, with the exception of sending for help. The only significant predictor of sending for help at 12-month follow-up was sending for help at baseline ($p = 0.001$).

**Table 4 Most common reasons for not providing first aid to a peer with a physical injury or illness.**

| Reason (n) | Example | Percentage of all responses (n = 158) | Percentage of codable responses (n = 109) | n (%) of codable responses from PFA group | n (%) of codable responses from tMHFA group |
|---|---|---|---|---|---|
| Someone else provided first aid (29) | 'The first aid team was already there' (P517, baseline) | 18.4 | 26.6 | 17 (58.6) | 12 (41.6) |
| Lacked adequate skills or experience (22) | 'Because I am not confident enough to do first aid' (P424, baseline) | 13.9 | 20.2 | 8 (36.4) | 14 (63.6) |
| Practical constraints (16) | 'Could not reach the person' (P1315, baseline) | 10.1 | 14.7 | 7 (43.7) | 9 (56.3) |
| Peer did not need first aid (11) | 'All they needed was a band-aid' (P1057, baseline) | 7.0 | 10.1 | 6 (54.5) | 5 (45.5) |
| Negative perspective of first aid or person needing first aid (10) | 'Too weirded out' (P184, baseline) | 6.3 | 9.2 | 5 (50.0) | 5 (50.0) |
| No reason (8) | 'There wasn't a reason' (P822, baseline) | 5.1 | 7.3 | 5 (62.5) | 3 (37.5) |
| Student is not sure why they didn't provide first aid (9) | 'Not sure' (P1591, baseline) | 5.7 | 8.3 | 4 (44.4) | 5 (55.6) |

Note:
A total of 115 responses were recorded at baseline and 43 at follow-up. Only categories with responses provided by more than five participants are included in this table. Other responses, reported by between one and five participants, included: being unaware that first aid was needed and no established rapport with the person. Logistic regressions that assessed whether reason for not providing PFA at baseline predicted reason for not providing PFA at follow-up (controlling for age, gender, English as first language, school, intervention received and number of first aid sessions attended) revealed no significant predictors.

from the methodology used to assess first aid responses. Other studies of school-based first aid programmes use knowledge or skills assessments to evaluate students' first aid competencies, while this research used an open-ended, free text response question. Many

responses were brief or lacked key details which would have enabled coders to assess the adequacy of students' first aid actions and assign a valid code. The resultant loss of data likely contributed to the null effects found in the analyses.

Analysis of the types of appropriate first aid provided to peers indicated that first aid was provided for injuries such as wounds, sprains/strains and fractures/dislocations. This highlights the importance of designing first aid courses for young people that enable them to address injuries and illnesses common to adolescence (*Reveruzzi, Buckley & Sheehan, 2016*). Adolescents who were unable to help a peer who needed first aid cited a range of reasons for this, including that others were already providing first aid (reported more often by PFA participants) and a belief that their skills and experience were not adequate (reported more often by tMHFA participants). It appears that only the reason related to lack of skill or experience has been reported previously in this age group (*Kanstad, Nilsen & Fredriksen, 2011*; *Ma et al., 2015*; *Omi et al., 2008*). These results should be regarded as preliminary as it is uncommon for studies to examine reasons for not providing first aid in response to a real-life emergency. Future research should attempt to replicate these findings with different, larger samples of adolescents and endeavour to understand these reasons in greater depth. Semi-structured interviews may offer insight into why adolescents with first aid training believe they lack the skills to provide first aid, or why they may feel reluctant to provide first aid to a peer (c.f. *Wilks et al. (2016)*, who found that primary school-aged children were less likely to feel confident helping a friend or same-aged stranger than a family member). This information could be used by course developers to address perceived barriers to helping among adolescents during training, and assess whether this affects rates of first aid provision.

Although further research is needed to better understand how first aid skills can best be acquired, retained and used outside of learning and testing environments, the results of this study suggest that schools can maximise the utility and relevance of school-based first aid programmes in two main ways. Firstly, schools can endeavour to ensure that their first aid programmes contain topics that are appropriate and useful for the target age group. The results of this research imply that the most commonly used first aid skills were performed in response to injuries and illnesses common to adolescence. Courses which incorporate content that reflects students' developmental stage alongside basic life support steps may increase the likelihood of young people applying first aid to a peer in an emergency, which may ultimately contribute to reductions in injury-related morbidity and mortality (*Tannvik, Bakke & Wisborg, 2012*). A second, complementary suggestion for schools is to consider introducing a first aid educational pathway into the curriculum that enables students to learn age-appropriate first aid skills during each school year (*De Buck et al., 2015*; *Wilks & Pendergast, 2017*). A first aid educational pathway, such as the one developed by *De Buck et al. (2015)* may serve many purposes, including revision of knowledge and skills from previous years alongside acquisition of new skills, improving students' confidence and competence in applying first aid, and improving teachers' confidence in teaching first aid to students of any age. The pathway could also be developed in consultation with students to ensure that it addresses topics and concerns

that are important to them, such as particular barriers to providing first aid (see Table 4). Embedding regular first aid lessons into the school curriculum, and including student input in their development, can help to normalise the subject and increase students' comfort, self-efficacy, knowledge and skills in this area (*Wilks & Pendergast, 2017*).

This research responds to calls in the first aid literature for more randomised controlled trials (*Van de Velde et al., 2009*), particularly ones assessing school-based first aid training and first aid behaviours (*Reveruzzi, Buckley & Sheehan, 2016*). The study used a rigorous cluster-randomised crossover trial methodology with longitudinal follow-up, and recruited a large, diverse sample of schools and adolescents. The structure and timing of each first aid intervention was matched to improve the likelihood that any changes to behaviour would be due to programme content and not an artefact of undertaking an activity outside of the usual curriculum. The content was designed to cover key first aid skills, such as CPR, and first aid injuries and emergencies common to adolescence, such as strains, sprains, wounds and anaphylaxis, to increase its relevance to, and utility for, the target audience (*De Buck et al., 2015*; *Reveruzzi, Buckley & Sheehan, 2016*). The findings also have ecological validity, as they provide some insight into whether and how first aid skills were used outside of learning and testing environments, and why they were not used.

Despite these advantages, limitations remain. Not including follow-up surveys between baseline and 12 month follow-up means that the research could not examine whether knowledge or motivation to assist were retained up to a point (e.g. 3 or 6 months after training; *Andresen et al., 2008*) before dropping off. As the questionnaire did not ask about prior first aid training at baseline, or whether students had undertaken additional first aid training between baseline and follow-up, the analyses could not control for these variables, or clarify whether any changes were due to the first aid training delivered as part of this research. The questionnaire also did not ask students what the problem requiring first aid was, and so it is not known whether the first aid provided was appropriate for the situation, or administered effectively; students' ability to correctly perform the first aid actions they were taught was not assessed. The small number of students responding to the follow-up survey, and the brief explanations of their first aid actions, precluded a more thorough assessment of first aid behaviour and reasons for not helping at 12-month follow-up. As the research relied on self-report and students' recollection of events that may have happened several months earlier, recall bias may have affected the quantity, quality and validity of the data. Lastly, a large proportion of open-ended responses could not be coded, which considerably reduced both the quantity and quality of data. Several of these limitations are being addressed in another large-scale trial comparing tMHFA and PFA which is currently underway, for example offering remuneration to increase participation rates, asking what the problem requiring first aid was and including questions about intentions and confidence to provide first aid to a peer experiencing different illnesses and injuries. It is anticipated that these changes will enhance the amount and quality of data on first aid actions performed by adolescents.

## CONCLUSIONS

The findings of this research highlight the need for more rigorously designed longitudinal studies to establish the effectiveness of school-based first aid programs, and more research that focuses on first aid behaviour in response to emergencies that adolescents may encounter. An improved understanding of first aid skill retention and use will contribute to the development of engaging and relevant health education for young people.

## ACKNOWLEDGEMENTS

The authors wish to thank Don Dumayas for developing the coding frame used to analyse the open-ended responses, Catherine Johnson and Penny Cropper for coding the responses and Betty Kitchener who assisted with the grant application which funded this research.

### Funding

This research was supported by a Mental Health Research grant awarded to the authors by Australian Rotary Health, and by a National Health and Medical Research Council Grant awarded to Anthony F. Jorm. The funders had no role in study design, data collection and analysis, decision to publish, or preparation of the manuscript.

### Grant Disclosures

The following grant information was disclosed by the authors:
Australian Rotary Health.
National Health and Medical Research Council.

### Competing Interests

Anthony F. Jorm and Amy J. Morgan are Academic Editors for PeerJ. Alyssia Rossetto is employed by Mental Health First Aid Australia as the Research Manager. Claire M. Kelly is employed by Mental Health First Aid Australia as the Director of Research and Curriculum. Anthony F. Jorm is a Director on the Board of the not-for-profit organisation Mental Health First Aid Australia that provides teen Mental Health First Aid instructor training to appropriately qualified individuals. No author will financially benefit from the results of this research.

### Author Contributions

- Alyssia Rossetto performed the experiments, analysed the data, prepared figures and/or tables, authored or reviewed drafts of the paper, and approved the final draft.
- Amy J. Morgan performed the experiments, analysed the data, prepared figures and/or tables, authored or reviewed drafts of the paper, and approved the final draft.
- Laura M. Hart conceived and designed the experiments, performed the experiments, analysed the data, authored or reviewed drafts of the paper, and approved the final draft.

- Claire M. Kelly conceived and designed the experiments, authored or reviewed drafts of the paper, and approved the final draft.
- Anthony F. Jorm conceived and designed the experiments, performed the experiments, analysed the data, prepared figures and/or tables, authored or reviewed drafts of the paper, and approved the final draft.

## Human Ethics

The following information was supplied relating to ethical approvals (i.e. approving body and any reference numbers):

Ethics approval was obtained from the University of Melbourne's Human Research Ethics Committee (HREC1341238) and the state government education department for Victoria (2014_002268).

## Clinical Trial Registration

The following information was supplied regarding Clinical Trial registration:

Australian New Zealand Clinical Trials Registry: ACTRN12614000061639, available from https://www.anzctr.org.au/Trial/Registration/TrialReview.aspx?id=365334&is Review=true.

## Data Availability

Raw data is available in the Supplemental Files.

## Supplemental Information

Supplemental information for this article can be found online at http://dx.doi.org/10.7717/peerj.9782#supplemental-information.

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
