# Peer review of "Frequency and quality of first aid offered by older adolescents: a cluster randomised crossover trial of school-based first aid courses"

_PeerJ, doi:10.7717/peerj.9782_

## Round 0.1 · original submission · Minor Revisions

· Academic Editor

Minor Revisions

Thank you for your submission. While there were some dependencies in the opinions of the reviewers, I am confident that with minor revisions your manuscript will meet the standards and expectations for publication.

Please clarify your, aim, research questions and hypothesis and make sure that your discussion section and conclusions clearly map back to them. This need not be a major revision, rather making sure the language is clear and concise.

Please see Reviewer One's suggestions as it relates to validity and findings. This I think will add clarity to the manuscript.

·

Basic reporting

Thank you for this interesting assessment of the effectiveness of school based first aid for adolescents. As a parent of adolescents who is interested in my children participating in a similar program I found this to be of particular interest. Similarly, I feel this topic will be of interest to school and other adolescent based programs who offer such opportunities.
Over all the manuscript is very well written and relatively easy to follow. The references, background and context provided are appropriate. The structure of the article, figures and tables are professional. One note,Table 1 needs a minor formatting change - line up the information in the columns in the first section (session 1).

One piece not included that you may wish to consider is the inclusion of your hypotheses.

Experimental design

The experiment design appears appropriate -albeit uncommon as you mixed both quantitative and qualitative methodology without really synthesizing the findings together. Further in the study design section you note that a cluster randomised crossover trial was conducted. I like how well matched the subjects were given the need to utilize more than one school. In this section however there isn't a mention of the qualitative design... aspects of the study or a provided linkage of how the two approaches complement one another. Also I think more detail regarding the coding framework used needs to be included. I would also like you to address the choice to quantify the qualitative data, as this approach is somewhat controversial among staunch qualitative researchers.

Measures: please speak to how the measures were created, who created them, and if validity or reliability analyses were carried out.

Validity of the findings

The validity of the findings are apparent. Consider adding some discussion as to the implications of the findings. Would schools / programs offer opportunities to refresh or practice skills at regular intervals? Should such first aid programs not be offered if findings bare out their ineffectiveness or does training offer something additional to teens such as feelings of confidence to be able to help in situations where a need arises? Perhaps a broader question needs to be answered. Other questions that come to mind are - 1. for CPR courses are the findings similar? What about life guard training or wilderness first aid? Are there regional or country differences noted. Does age of participant matter? Perhaps year 10 is the wrong target year.
Just a little fuel for thought as you seek to continue this work and widen the understanding of this topic.

Additional comments

Well done - nicely written piece. Enjoyable to read.

·

Basic reporting

This is a very professional piece of research exploring an important topic for adolescents. The paper is extremely well written and adequately covers the relevant literature. The structure conforms to the journal standards and all basic reporting criteria are met.

Experimental design

The experimental design is rigorous and responds to identified gaps in the field previously noted by He et al (2014) and Reveruzzi et al (2016). The methods are described in sufficient detail for replication and the supplementary files provide very clear guidance for other researchers. In passing, lines 68-70 of the Introduction have text that is replicated (printing error).

He Z, Wynn P & Kendrick D (2014) Non-resuscitative first-aid training for children and laypeople: A systematic review. Emergency Medicine Journal, 31: 763-768

Reveruzzi B, Buckley L & Sheehan M (2016) School-based first aid training programs: A systematic review. Journal of School Health, 86: 266-272

Validity of the findings

I especially enjoyed the rigorous methodology employed in this study. In the end the results were not surprising and in keeping with earlier studies – no differences between the groups on the frequency of appropriate first aid actions performed 12 months after training and the most common actions being sending for help and wound care. Apart from the rigorous methodology employed the other real value of the paper is in the discussion where the authors have looked at possible limitations such as using open ended free text responses in an online survey . The authors identify where improvements can be made to investigate this topic and have indicated that a current study addresses these points. A particularly interesting topic would be to investigate willingness to offer first aid and CPR to family, friends and strangers and which barriers operate to reduce the likelihood of offering help to each group (fear of making a mistake and hurting the patient, legal disputes, fear of infection). These topics have previously been discussed in the literature, with greater willingness to assist family, then friends, then same age strangers and lastly adult strangers (Wilks et al., 2016). The distinction between assisting same age friends and same age strangers has important implications for real world adolescent bystander responses.

Overall, this is a very good paper that makes a very worthwhile contribution to the literature.

Wilks J, Kanasa H, Pendergast D & Clark K (2016) Emergency response readiness for primary school students. Australian Health Review, 40: 357–363

Additional comments

This is a very good paper that makes a very worthwhile contribution to the literature.

·

Basic reporting

Trial Registration- The ethical trial registration lists a study of performing first aid versus intervening during a mental health crisis based on the training provided to the cohorts. Whereas the abstract discusses the impacts of physical first aid training versus teen mental health first aid in the frequency of performing these skills on their peers. Was ANZCTR advised of the change?
From Abstract-“This study aimed to establish whether the training programs produced differences in the frequency with which students offered and provided first aid to peers experiencing an illness or injury and the amount of first aid provided that was in line with current first aid guidelines.”
From Ethical Applications- “Cluster randomised controlled trial of secondary school student training in teen Mental Health First Aid versus physical first aid on ability to assist peers with a mental health problem.”

In general the article is well written with professional English and structure.

Some clarity would be helpful in regards to the original hypotheses whether the subjects are responding to physical or emotional emergencies.

Experimental design

As listed above, it was difficult to correlate the results based on the difference between the initial goals and the results.

Validity of the findings

Unable to validate until clarification is given on the research question and hypothesis, as I feel that these may change.

Additional comments

I don't feel that I am able to get to the point of verifying the accuracy and references and so on until I receive clarification on the desired results. I feel that the manuscript is very hard to evaluate as I am continuously having to question the true aim of the study.

---

## Round 0.2 · accepted · Accept

· Academic Editor

Accept

Thank you for addressing the concerns and request for clarification throughout your manuscript. I do understand the the publication process is iterative and appreciate your commitment to the process